# Dietary Supplementation of Aspirin Promotes *Drosophila* Defense against Viral Infection

**DOI:** 10.3390/molecules28145300

**Published:** 2023-07-09

**Authors:** Fanrui Kong, Abdul Qadeer, Yali Xie, Yiheng Jin, Qingyang Li, Yihua Xiao, Kan She, Xianrui Zheng, Jiashu Li, Shanming Ji, Yangyang Zhu

**Affiliations:** 1Center for Developmental Biology, School of Life Sciences, Anhui Agricultural University, Hefei 230036, Anhui, China; kongfanrui@stu.ahau.edu.cn (F.K.); abdulqadeer@atu.ahau.edu.cn (A.Q.); xieyali2002@163.com (Y.X.); jinyh@stu.ahau.edu.cn (Y.J.); qingyangli0312@126.com (Q.L.); west1130@163.com (Y.X.); shekan@stu.ahau.edu.cn (K.S.); 2Zhangzhou Affiliated Hospital of Fujian Medical University, Zhangzhou 363000, Fujian, China; xrzheng018@126.com; 3Université de Strasbourg, 67000 Strasbourg, France; heaidelijiashu@gmail.com

**Keywords:** aspirin, STING signaling, IMD pathway, antiviral immunity, *Drosophila*

## Abstract

Aspirin, also known as acetylsalicylic acid, is widely consumed as a pain reliever and an anti-inflammatory as well as anti-platelet agent. Recently, our studies using the animal model of *Drosophila* demonstrated that the dietary supplementation of aspirin renovates age-onset intestinal dysfunction and delays organismal aging. Nevertheless, it remains probable that aspirin plays functional roles in other biological activities, for instance antiviral defense reactions. Intriguingly, we observed that the replications of several types of viruses were drastically antagonized in *Drosophila* macrophage-like S2 cells with the addition of aspirin. Further in vivo experimental approaches illustrate that adult flies consuming aspirin harbor higher resistances to viral infections with respect to flies without aspirin treatment. Mechanistically, aspirin positively contributes to the *Drosophila* antiviral defense largely through mediating the STING (stimulator of interferon genes) but not the IMD (immune deficiency) signaling pathway. Collectively, our studies uncover a novel biological function of aspirin in modulating *Drosophila* antiviral immunity and provide theoretical bases for exploring new antiviral treatments in clinical trials.

## 1. Introduction

Viruses are a type of microorganism that may have a wide range of negative effects on the human body including damages to tissues/organs, immune system dysfunctions, respiratory illnesses, neurological symptoms, and increased risks of cancers. In order to prevent or treat viral infections, a large body of efforts have been dedicated to the field of “exploring antiviral drugs”, which was obviously one of the hottest scientific topics during the global pandemic of COVID-19 (caused by the SARS-CoV-2 virus). In general, antiviral drugs function by targeting specific parts of viruses, thereby preventing them from reproducing and spreading throughout the body. When taken as directed, these drugs can help to reduce the severity and duration of viral illnesses and prevent complications (for details, please see reviews including [1,2,3,4,5,6,7]). To date, there has been a significant breakthrough in the development of antiviral drugs, as many of them, including nucleoside inhibitors, protease inhibitors, neuraminidase inhibitors, entry inhibitors, RNA polymerase inhibitors, and integrase inhibitors, have been successively adopted in clinical trials. Nevertheless, antiviral drugs are not always a promise due to the facts that (1) viruses are notoriously difficult to combat as they are able to mutate quickly and develop resistance to drugs; (2) novel viruses, for instance SARS-CoV-2, regularly emerge and cause outbreaks, but existing antiviral drugs are ineffective against them. Therefore, exploring new antiviral drugs/compounds is crucial for responding to the ongoing threats posed by viral diseases.

The fruit fly (*Drosophila melanogaster*) has long been used as an animal model for studying a wide range of biological processes and diseases, including viral infections. In recent decades, there have been growing interests in utilizing *Drosophila* for identifying and testing potential antiviral drugs [8,9]. One advantage of using *Drosophila* as a model organism is that *Drosophila* has a relatively simple and well-characterized immune system, which makes it quite easy to study the interactions between viruses and host cells. In addition, *Drosophila* is a type of cost-effective and easy-to-maintain insect that can be used in large-scale screening assays in order to identify potential antiviral compounds [10,11]. For instance, a study by Adamson and colleagues established a *Drosophila* model of influenza virus infection by stably expressing the *M2* gene in specific tissues/organs [12]. They confirmed the antiviral efficiency of amantadine in restricting influenza titers, supporting the probable use of this system for high-throughput antiviral drug screening [12]. Another approach of infecting flies with HSV (herpes simplex virus) identified potential host factors that could be targeted by antiviral drugs [13]. Moreover, Lee and colleagues lately succeeded in expressing the *Orf6* (*open reading frame 6*) gene of SARS-CoV-2 in fly tissues, which displayed corresponding disease phenotypes [14], making it even more promising to explore functional materials against COVID-19. Overall, the use of *Drosophila* as a model organism for exploring antiviral drugs is an active area of scientific research, and has the potential to identify new compounds and targets for treating viral infections.

Aspirin (acetylsalicylic acid), a type of well-known NSAID (nonsteroidal anti-inflammatory drug), is commonly consumed by patients in order to relieve pain, reduce inflammation, and lower fever (reviewed in [15,16]). Although it is not typically used as an antiviral drug, recent in vitro studies have shown that it may inhibit the replication of several types of viruses, including HCV (hepatitis C virus), influenza virus, HSV (herpes simplex virus), and HIV (human immunodeficiency virus) [17,18,19]. However, it remains largely unknown whether or not aspirin executes antiviral activity in vivo. We lately reported the beneficial effect of aspirin on preventing age-onset gut epithelial dysfunction and delaying aging in *Drosophila* [20]. Encouraged by this, our next objective is to explore the potential antiviral role of aspirin by utilizing a fruit fly animal model.

In the present study, we provide in vivo evidence displaying that the dietary consumption of aspirin positively contributes to the fly antiviral defense against various types of viruses. Of interest is that aspirin likely mediates the STING (stimulator of interferon genes) but not the IMD (immune deficiency) signaling pathway to exert its essential role in the host defense against viral infections. Collectively, our studies shed light on the physiological function of aspirin in antiviral immunity and provide bases for the clinical usage of aspirin as an antiviral drug/treatment.

## 2. Results

### 2.1. Additional Aspirin Leads to Decreased Viral Titers in Drosophila S2 Cells

We first carried out experiments on *Drosophila* macrophage-like S2 cells to confirm the antiviral function of aspirin. Cultured S2 cells were pretreated with aspirin at various concentrations (0.1 and 0.5 mM, respectively), followed by infection from DCV (*Drosophila* C virus), which is one of the widely utilized viruses to infect S2 cells. As illustrated in Figure 1A, the viral titer of DCV was drastically decreased in S2 cells with additional aspirin (0.1 mM) compared to that in the control group. A more significant trend was observed when S2 cells were treated with a higher dose of aspirin (0.5 mM) (Figure 1A). These results suggested that aspirin prevents viral replication in a concentration-dependent manner. We next examined the impact of aspirin on the loads of other types of viruses. A strong decrease in CrPV (cricket paralysis virus) RNA was noticed in S2 cells with the treatment of aspirin (Figure 1B). A similar effect was also detected when S2 cells were infected with another RNA virus, VSV (vesicular stomatitis virus) (Figure 1C). However, additional aspirin hardly affected the titers of either SINV (sindbis virus) (Figure 1D) or DNA virus IIV6 (invertebrate iridescent virus 6) (Figure 1E). Taken together, our data support the notion that aspirin antagonizes viral replication in cultured *Drosophila* S2 cells largely depending on the specific type of virus.

### 2.2. Viral Proliferation Is Limited in Adult Flies with Dietary Supplementation of Aspirin

Encouraged by this, we then sought to illustrate whether or not aspirin executes antiviral activity in vivo. To determine this, we first raised the *wild-type* (*w^1118^* or *Canton-S*) flies with a medium supplemented with different concentrations of aspirin (0, 0.5, and 1 mg/L, respectively) and the dietary addition of aspirin hardly affected the food consumption of tested flies [20]. We then infected these flies with DCV and detected the viral titer on the following days (day 0, 1, and 2). As demonstrated in Figure 2A and Appendix A, on day 1 after infection, the DCV RNA levels were markedly reduced in the flies consuming 0.5 mg/L of aspirin with respect to those in the control group. A greater reduction was detected when flies were supplemented with a higher dose of aspirin (1 mg/L) (Figure 2A and Appendix A). Consistently, similar results were obtained when flies were collected for monitoring the viral titers on day 2 after infection (Figure 2A and Appendix A), implying that aspirin harbors antiviral function in adult flies. To further confirm these observations, we infected *w^1118^* males with other types of viruses, including CrPV and VSV. Intriguingly, the replications of both viruses were limited in the host flies by aspirin in a dosage-dependent manner (Figure 2B,C).

### 2.3. Dietary Supplementation of Aspirin Enhances Fly Survival after Viral Infection

Since the viral load was decreased in flies consuming aspirin, one would expect the stronger resistance of these flies against viral infection. Indeed, when we conducted survival rate assays after viral infections, we observed an increased viability of flies with the dietary addition of aspirin, compared to that of the controls (Figure 2D,G).

In summary, our data indicate that aspirin behaves as an antiviral compound in the fly defense against viral infection.

### 2.4. Aspirin Modulates Fly Survival and Viral Titers after Infection in an IMD-Independent Manner

We then aimed to illustrate how aspirin benefits the antiviral defense of *Drosophila*. Based upon our recent finding that aspirin delays fly aging via targeting Imd [20], which is a key adaptor protein in the IMD (immune deficiency) innate immune pathway [21], and the fact that the IMD signal plays an essential role in *Drosophila* antiviral immunity [22], we would like to hypothesize that aspirin executes an antiviral function via mediating Imd. In this regard, we examined the antiviral role of aspirin in the *imd* LOF (loss of function) mutant flies. Unexpectedly, the dietary supplementation of aspirin still restricted the replication of injected DCV (Figure 3A) and promoted the survival of the *imd* LOF mutants (Figure 3B). These results implied that *imd* is largely not responsible for aspirin positively contributing to the fly antiviral defense.

### 2.5. Aspirin Relies on Sting and Relish to Regulate Fly Survival and Viral Replication after Infection

Pioneering studies have delineated that in addition to the IMD signaling cascade, the Toll [23], JAK-STAT (Janus kinase-signal transducer and activator of transcription) [24], RNAi (RNA interference) [25], autophagy [26], and STING (stimulator of interferon genes) [27] pathways are also involved in governing the antiviral immune defense system of *Drosophila*. We then sought to identify whether or not aspirin mediates one (or some) of them to unleash its antiviral activity. As shown in Figure 3C–J, the *dif* (*dorsal-related immunity factor*), *hop* (*hopscotch*), *ago2* (*argonaute 2*), or *atg7* (*autophagy-related 7*) LOF mutant flies supplemented with aspirin displayed decreased mortalities. The DCV loads in these flies were reduced after viral injection (Figure 3C,J). However, when we dietarily supplemented the *sting* LOF mutants with aspirin, we observed that aspirin almost failed to affect fly survival and the viral titer upon DCV infection (Figure 4A,B). Moreover, we obtained similar results when we performed these assays utilizing the *rel* (*relish*) LOF mutant flies (Figure 4C,D). Collectively, our data indicated that aspirin benefits the fly antiviral defense largely in a STING-NF-kB (nuclear factor kappa B) signal-dependent manner.

### 2.6. Additional Aspirin Increases the Expressions of STING-Induced Genes

To obtain more evidence, we examined whether or not the expressions of STING downstream antiviral effectors are modulated by aspirin. When we infected the *wild-type* flies with DCV under normal rearing conditions, we detected enormous inductions of *nazo* and *srg1* (*STING-regulated gene 1*) (Figure 4E,F), which were consistent with previous findings [27,28]. Of interest is that the dietary supplementation of aspirin drastically increased the transcript levels of both *nazo* and *srg1*, but this was not the case in the *sting* LOF mutants (Figure 4E,F). These results indicate that aspirin positively participates in *Drosophila* antiviral defense reactions through mediating the STING-NF-kB signaling cascade.

## 3. Discussion

In the present study, we conducted both in vitro and in vivo experimental approaches to identify the antiviral role of aspirin. We provided compelling evidence showing that the dietary supplementation of aspirin not only lowers the mortality rate of *Drosophila* upon the infection of specific viruses, but also prevents the viral loads in the host flies. Of note is that the loss of *sting* or *rel* reverses the advantageous assessment of aspirin in the fly antiviral defense, implying that aspirin benefits *Drosophila* antiviral immunity in a STING-NF-kB-dependent manner. Thus, our studies demonstrate the functional role of aspirin against viral infection in vivo.

Aspirin has long been well-known for its anti-inflammation and anti-platelet function, and widely utilized in medical treatments [15,16]. It was lately shown to harbor an antiviral function [17,18,19,29], but these studies were mostly conducted in *vitro* systems. As a result, the prior objective of this research was to explore the potential involvement of aspirin in modulating the host antiviral defense reaction in vivo. To address this issue, we paid attention to the insect *Drosophila*, for which we have already succeeded in establishing the model of dietary consumption of aspirin and illustrating the anti-aging role of aspirin [20]. As expected, the dietary supplementation of aspirin significantly enhanced the expressions of antiviral effectors and the survival of flies upon viral infections. In this regard, our studies not only confirm the antiviral role of aspirin that was suggested in the previous literature [17,18,19,29], but also provide for the first-time in vivo evidence to support this notion. Of interest is that aspirin enables the antagonization of several types of viruses, making it possible to become a broad antiviral drug in clinical medication.

How does aspirin benefit the *Drosophila* antiviral defense? Our recent findings illustrated that aspirin down-regulates the K63 (63rd lysine)-linked ubiquitination of the Imd protein, thereby contributing to gut immune homeostasis and epithelial function [20]. Therefore, we first explored whether or not aspirin is involved in the fly antiviral reaction via modulating Imd. Unexpectedly, dietary aspirin supplementation still enhanced the antiviral immunity in the *imd* LOF mutates. These observations encouraged us to further examine the antiviral role of aspirin in flies including *dif*, *hop*, *ago2*, *atg7*, or *sting* LOF mutants, where the known antiviral pathways (Toll, Jak-STAT, RNAi, autophagy, and STING-NF-κB, respectively) were blocked separately. Intriguingly, the STING-NF-κB signal turned out to be responsible for the antiviral activity of aspirin in adult flies. Nevertheless, knowledge regarding how aspirin modulates the STING-NF-κB signaling pathway is still lacking. Based upon our experience and knowledge with regard to the molecular mechanism by which aspirin can mediate protein ubiquitination, one mechanism we would like to propose is that the ubiquitination modification of a (some) key factor(s) in the STING-NF-κB signaling cascades is (are) regulated by aspirin.

Another question that needs to be addressed is that on which tissue/organ aspirin exerts its antiviral effect. Since we raised experimental flies with the dietary supplementation of aspirin, it is probable that aspirin functions in the digestive system to enhance fly metabolism, thereby contributing to the host antiviral response. In agreement with this, a previous transcriptomic analysis showed that aspirin can alter the expression of a series of genes involved in the metabolic pathway [30]. On the other hand, aspirin may circulate into fat body (the main immune tissue/organ during systemic infection), where it upregulates STING-NF-κB activity. It would thus be worthwhile in the future to explore (1) the potential regulatory relationship between aspirin and the Sting protein; (2) the antiviral function of aspirin in a tissue/organ-specific manner, for instance with the use of flies with the gut or fat body-specific RNAi of *sting*.

## 4. Significance and Limitations of this Study

Our studies uncover the functional involvement of aspirin in the fly defense against several types of viruses. Even though the genetic evidence implies an intersection between aspirin and the STING-NF-κB signaling pathway, the underlying molecular mechanism is still not fully addressed. Further experimental approaches at the levels of molecular biology and biochemistry are required to identify the substrate of aspirin and to illustrate how aspirin modulates the *Drosophila* STING-NF-κB signaling cascades.

## 5. Materials and Methods

### 5.1. Fly Strains and Husbandry

All flies in this study were raised under normal conditions (25 °C, 12 h/12 h light/dark cycle, and 60% humidity). The detailed information on flies in this study is as follows: (1) *atg7^d4^* (#95244), *hop^3^* (#8495), *hop^27^* (#8493), and *rel^E20^* (#55714) were purchased from Bloomington *Drosophila* Stock Center; (2) the *ago2^414^*, *dif^1^*, *IMD^1^*, and *STING^Rxn^* strains were described previously [27,28,31,32].

To prepare the fly medium supplemented with aspirin, we first dissolved aspirin (#A2093, Sigma-Aldrich, St. Louis, USA) in sterile H_2_O (37 °C) at final concentrations of 0.5 and 1 g/L. Then, 1 mL of each aspirin solution was added into 1 L of the fly medium to prepare food with 0.5 or 1 mg/L of aspirin.

### 5.2. S2 Cell Manipulation

S2 cells were cultured in an insect medium (Hyclone, Amersham, UK) supplemented with 10% FBS (fetal bovine serum, Hyclone) and Penicillin/Streptomycin (Invitrogen, Carlsbad, USA). The MycoAlert^TM^ kit (Lonza, Basel, Switzerland) was used for the mycoplasma detection and validation of the authenticity of S2 cells. For aspirin treatment in S2 cells, we first prepared a 5 and 10 mM aspirin solution (dissolved in sterile H_2_O). Then, 0.1 mL of each solution was added into S2 cells (0.9 mL) to reach the final concentration of aspirin, this being 0.5 or 1 mM, respectively. An equal volume of sterile H_2_O was added in the control group. Six hours after aspirin treatment, S2 cells were infected with DCV (MOI 1), CrPV (MOI 0.1), FHV (MOI 1), VSV (MOI 1), or IIV6 (MOI 1).

### 5.3. Viral Infection in Flies and Survival Rate Assays

Male flies (3 to 5 days old) were injected at the thorax with 4.6 nL of the viral particle solution (500 pfu per fly for DCV and VSV; 50 pfu per fly for CrPV) using a nanoliter injector (Nanoject III, Drummond, Broomall, USA). An equal volume of Tris-HCl solution (10 mM, pH = 7.5) was injected in the control group. Flies were then counted daily for survival. Flies that (<10%) died within 2 h were not considered in this assay.

### 5.4. RT-qPCR Assays

For cell samples, S2 cells were harvested and directly lysed in Trizol reagent (Invitrogen). For fly samples, animals were collected and homogenized in Trizol with glass beads. Total RNA was extracted using the standard chloroform–isopropanol method and the quality of RNA was examined via agarose gel electrophoresis. The reverse transcription kit (Transgen, Beijing, China) was used for cDNA synthesis and quantitative PCR assays were performed by using SYBR reagents (Invitrogen) [33]. Primers used in these experiments are outlined below. RpL32-s: GCCGCTTCAAGGGACAGTATCT; RpL32-as: AAACGCGGTTCTGCATGAT; DCV-s: TCATCGGTATGCACATTGCT; DCV-as: CGCATAACCATGCTCTTCTG; CrPV-s: GCTGAAACGTTCAACGCATA; CrPV-as: CCACTTGCTCCATTTGGTTT; FHV-s: TTTAGAGCACATGCGTCCAG; FHV-as: CGCTCACTTTCTTCGGGTTA; VSV-s: CATGATCCTGCTCTTCGTCA; VSV-as: TGCAAGCCCGGTATCTTATC; IIV6-s: TTGTTAGGAATTGGAACTGGAA; IIV6-as: GCCCTAGATGCTGCTTGTTC.

### 5.5. Statistical Analyses

All statistical analyses were performed using GraphPad Prism 8. The statistical significance in Figure 1A,E, Figure 2A,C, Figure 3A,C,E,G,I, Figure 4A,C,E and Appendix A was determined using the ANOVA or Mann–Whitney tests. In Figure 2D,F, Figure 3B,D,F,H,J and Figure 4B,D,F, the log-rank test (Kaplan–Meier method) was used for statistical analysis. A *p* value of less than 0.05 was considered statistically significant. * indicates *p* < 0.05; ** indicates *p* < 0.01; *** indicates *p* < 0.001; ns indicates non-significance.

## Figures and Tables

**Figure 1 molecules-28-05300-f001:**
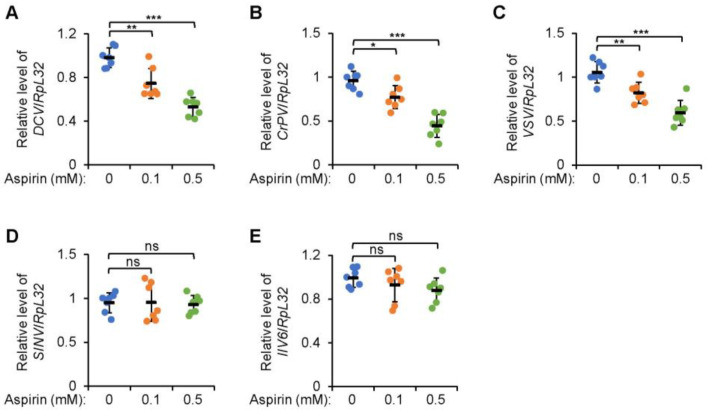
Aspirin prevents viral replication in *Drosophila* S2 cells. (**A**–**E**) *Drosophila* S2 cells were pretreated with aspirin (final concentration at 0, 0.1, and 0.5 mM, respectively) for 6 h. Cells were infected with viruses including DCV (**A**), CrPV (**B**), VSV (**C**), SINV (**D**), and IIV6 (**E**) as indicated. 24 h after viral infection, cells were lysed for viral titer examination. Each dot represents one independent replicate and data are shown as mean plus standard errors. *, *p* < 0.05; **, *p* < 0.01; ***, *p* < 0.001; ns, not significant.

**Figure 2 molecules-28-05300-f002:**
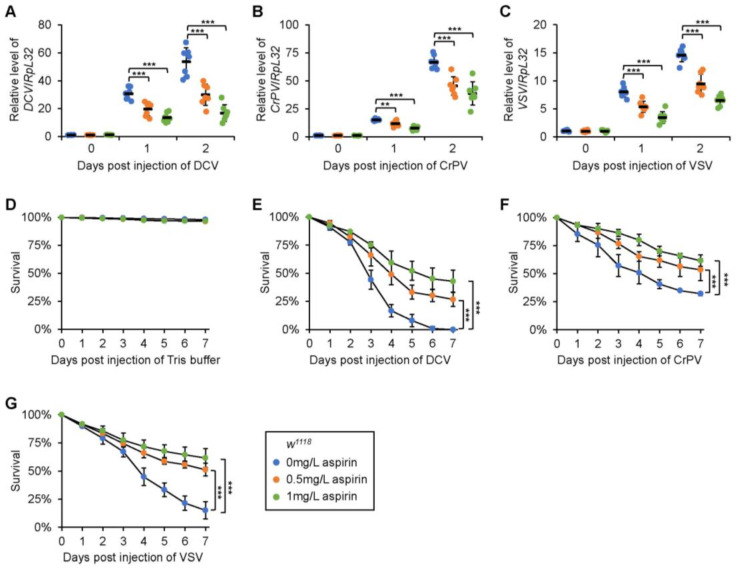
Dietary supplementation of aspirin promotes antiviral immunity in *Drosophila*. (**A**–**C**) Fresh male progenies of *w^1118^* flies were collected and raised on the medium supplemented with aspirin (0, 0.5, and 1 mg/L, respectively). At the age of 3 to 5 days, flies were infected with viruses including DCV (**A**), CrPV (**B**), and VSV (**C**). At indicated time points (day 0, 1, and 2), flies were collected to monitor the viral loads. Each dot represents one independent replicate (5 flies) and data are shown as mean plus standard errors. (**D**–**G**) Male *w^1118^* flies were raised and infected with Tris-HCl buffer (**D**) or viruses (**E**–**G**) similarly to the procedures followed in (**A**–**C**), followed by survival rate analyses. The numbers of flies are as follows. In (**D**), *w^1118^* flies without aspirin: 106, 105, and 104; *w^1118^* flies supplemented with 0.5 mg/L of aspirin: 104, 105, and 106; *w^1118^* flies supplemented with 1 mg/L of aspirin: 105, 107, and 103. In (**E**), *w^1118^* flies without aspirin: 102, 104, and 105; *w^1118^* flies supplemented with 0.5 mg/L of aspirin: 104, 104, and 102; *w^1118^* flies supplemented with 1 mg/L of aspirin: 105, 106, and 104. In (**F**), *w^1118^* flies without aspirin: 105, 104, and 106; *w^1118^* flies supplemented with 0.5 mg/L of aspirin: 102, 103, and 104; *w^1118^* flies supplemented with 1 mg/L of aspirin: 104, 103, and 107. In (**G**), *w^1118^* flies without aspirin: 104, 105, and 103; *w^1118^* flies supplemented with 0.5 mg/L of aspirin: 103, 104, and 107; *w^1118^* flies supplemented with 1 mg/L of aspirin: 106, 105, and 104. In (**A**–**C**) and (**E**–**G**), **, *p* < 0.01; ***, *p* < 0.001.

**Figure 3 molecules-28-05300-f003:**
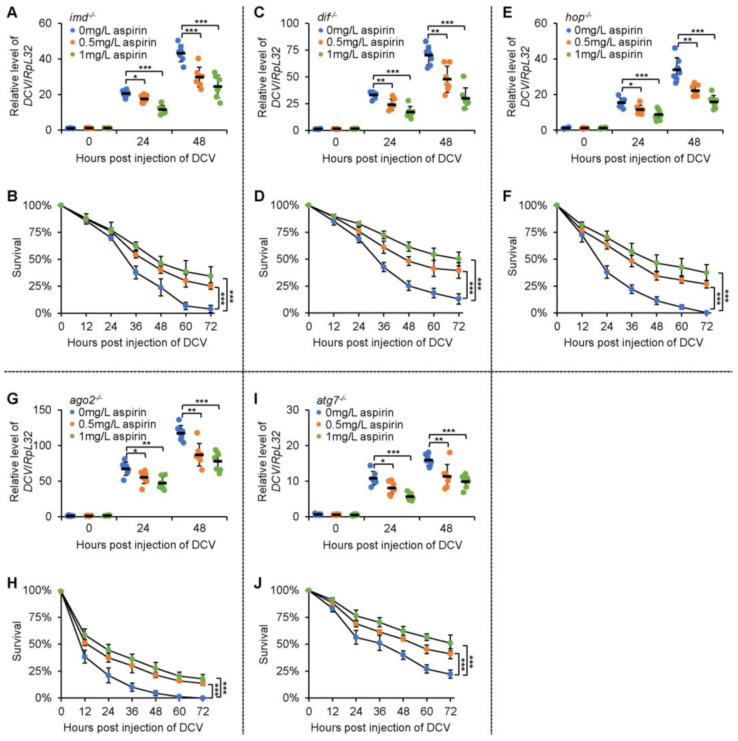
Aspirin enhances the antiviral defenses of several types of mutant flies. (**A**,**B**) The *imd* LOF mutants were raised in a medium supplemented with or without aspirin as indicated. Flies were infected with DCV and subjected to viral titer detection at various time points (**A**) or survival assays (**B**). (**C**–**J**) The *dif* (**C**,**D**), *hop* (**E**,**F**), *ago2* (**G**,**H**), and *atg7* (**I**,**J**) LOF mutants were subjected to similar experiments to those in (**A**,**B**). In (**A**,**C**,**E**,**G**,**I**), each dot represents one independent replicate (5 flies) and data are shown as mean plus standard errors. The numbers of flies for survival assays are as follows. In (**B**), *imd^−/−^* flies without aspirin: 102, 104, and 104; *imd^−/−^* flies supplemented with 0.5 mg/L of aspirin: 102, 105, and 106; *imd^−/−^* flies supplemented with 1 mg/L of aspirin: 104, 103, and 104. In (**D**), *dif^−/−^* flies without aspirin: 104, 103, and 105; *dif^−/−^* flies supplemented with 0.5 mg/L of aspirin: 102, 104, and 103; *dif^−/−^* flies supplemented with 1 mg/L of aspirin: 102, 106, and 104. In (**F**), *hop^−/−^* flies without aspirin: 105, 102, and 106; *hop^−/−^* flies supplemented with 0.5 mg/L of aspirin: 105, 105, and 103; *hop^−/−^* flies supplemented with 1 mg/L of aspirin: 103, 104, and 105. In (H), *ago2^−/−^* flies without aspirin: 101, 105, and 103; *ago2^−/−^* flies supplemented with 0.5 mg/L of aspirin: 104, 103, and 103; *ago2^−/−^* flies supplemented with 1 mg/L of aspirin: 103, 102, and 104. In (J), *atg7^−/−^* flies without aspirin: 105, 104, and 102; *atg7^−/−^* flies supplemented with 0.5 mg/L of aspirin: 103, 104, and 104; *atg7^−/−^* flies supplemented with 1 mg/L of aspirin: 102, 106, and 105. In (**A**–**J**), *, *p* < 0.05; **, *p* < 0.01; ***, *p* < 0.001.

**Figure 4 molecules-28-05300-f004:**
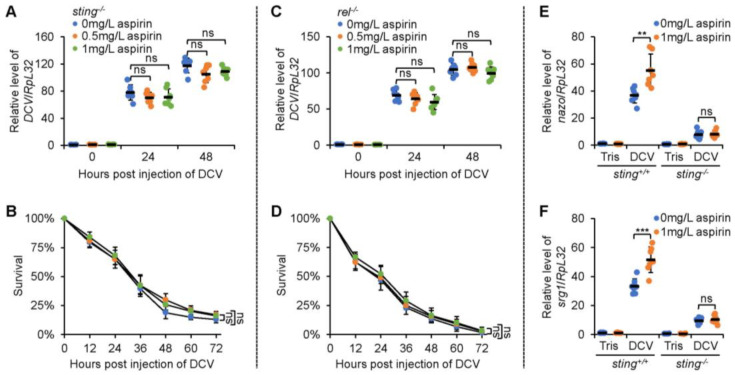
Loss of *sting* or *rel* prevents the antiviral activity of aspirin. (**A**,**B**) The *sting* LOF mutants were raised in medium supplemented with or without aspirin as indicated. Flies were infected with DCV and subjected to viral titer detection at various time points (**A**) or survival assays (**B**). (**C**,**D**) The *rel* LOF mutants were subjected to similar experiments to those in (**A**,**B**). (**E**,**F**) The *sting* LOF mutants and control flies were raised in a medium supplemented with or without aspirin. Flies were injected with Tris-HCl buffer or DCV as indicated, followed by RT-qPCR assays to monitor the expression levels of *nazo* (**E**) and *srg1* (**F**). In (**A**,**C**,**E**,**F**), each dot represents one independent replicate (5 flies) and data are shown as mean plus standard errors. The numbers of flies for survival assays are as follows. In (**B**), *sting^−/−^* flies without aspirin: 101, 104, and 104; *sting^−/−^* flies supplemented with 0.5 mg/L of aspirin: 104, 105, and 102; *sting^−/−^* flies supplemented with 1 mg/L of aspirin: 103, 102, and 101. In (**D**), *rel^−/−^* flies without aspirin: 105, 103, and 102; *rel^−/−^* flies supplemented with 0.5 mg/L of aspirin: 102, 106, and 103; *rel^−/−^* flies supplemented with 1 mg/L of aspirin: 101, 102, and 104. In (**A**–**F**), **, *p* < 0.01; ***, *p* < 0.001; ns, not significant.

## Data Availability

The data supporting the findings of this study are available from the corresponding author upon reasonable request.

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
