# Peer review of "Dietary Supplementation of Aspirin Promotes *Drosophila* Defense against Viral Infection"

_molecules, 2023, doi:10.3390/molecules28145300_

Round 1

Reviewer 1 Report

The manuscript titled "Dietary supplementation of aspirin promotes Drosophila defense against viral infection" explores the antiviral effects of aspirin in Drosophila. The study demonstrates that dietary supplementation of aspirin inhibits viral replication in Drosophila macrophage-like cells and adult flies, leading to increased resistance against viral infections. Mechanistically, aspirin mediates the STING signaling pathway rather than the IMD pathway to enhance the fly's antiviral defense. The findings shed light on a novel biological function of aspirin in modulating antiviral immunity and suggest its potential clinical usage as an antiviral treatment. The study primarily focuses on the effects of aspirin on viral replication and resistance, and further research is needed to elucidate the potential side effects or adverse reactions associated with aspirin supplementation. To improve the manuscript for acceptance in the journal, several corrections are required.

Comments to improve the manuscript:

To improve the manuscript for acceptance in the journal, rewrite the abstract to provide a concise summary of the study's objectives, methods, results, and significance, while ensuring the introduction offers a comprehensive overview of the current landscape of antiviral drug research, including recent advancements and challenges. Furthermore, enhance the methods section by providing more details on the experimental procedures, and revise the figures for better clarity and readability. Finally, expand the discussion section to discuss the implications of the findings and potential future directions, such as exploring the effects of aspirin on other viral pathogens and investigating its safety and efficacy in human models.

1. Abstract

a. In the abstract, provide a clear statement about the significance and novelty of the study findings.

b. Consider rephrasing the sentence "Nevertheless, it remains probable that aspirin enables to execute other undescribed biological functions" to a more concise and direct statement.

2. Introduction

a. Provide a more focused and concise introduction to the topic of antiviral drugs and the need for new compounds/treatments.

b. Consider reorganizing the information to present a logical flow of ideas and concepts.

c. Provide a clear research gap and highlight the specific contribution of the current study in addressing this gap.

3. Results

a. Subdivide the results section into subsections for clarity. Use more descriptive subheadings for each subsection.

b. Provide more context and explanation for the experimental methods used in S2 cell and fly experiments.

c. Clarify the abbreviations "DCV", "CrPV", "VSV", "SINV", and "IIV6" and provide their full name upon first mention.

d. Provide statistical analysis details and specify the significance thresholds for the results presented in Figure 1.

e. Provide a more detailed explanation of the survival rate analyses presented in Figures 2D-2G.

f. Provide statistical analysis details and specify the significance thresholds for the results presented in Figures 2 and 3.

4. Discussion

a. Discuss the potential mechanisms underlying the antiviral effects of aspirin in more detail.

b. Compare and contrast the findings of the current study with previous studies on the antiviral effects of aspirin.

c. Address the limitations of the study and suggest future directions for research.

5. Overall

Improve the clarity and organization of the manuscript by rephrasing and restructuring sentences and paragraphs. Proofread the manuscript for grammatical and typographical errors. Ensure consistency in terminology and abbreviations throughout the manuscript.

Moderate editing of English language required

Author Response

The manuscript titled "Dietary supplementation of aspirin promotes Drosophila defense against viral infection" explores the antiviral effects of aspirin in Drosophila. The study demonstrates that dietary supplementation of aspirin inhibits viral replication in Drosophila macrophage-like cells and adult flies, leading to increased resistance against viral infections. Mechanistically, aspirin mediates the STING signaling pathway rather than the IMD pathway to enhance the fly's antiviral defense. The findings shed light on a novel biological function of aspirin in modulating antiviral immunity and suggest its potential clinical usage as an antiviral treatment. The study primarily focuses on the effects of aspirin on viral replication and resistance, and further research is needed to elucidate the potential side effects or adverse reactions associated with aspirin supplementation. To improve the manuscript for acceptance in the journal, several corrections are required.

    Response: We are very much grateful for the reviewer spending time evaluating our manuscript and positively commenting on our work. We deeply appreciate the reviewer’s constructive comments and suggestions which help us to further improve the quality of our paper. The detailed point-by-point responses to reviewer’s comments are outlined below.

In the abstract, provide a clear statement about the significance and novelty of the study findings.

    Response: In the Section of Abstract, we have included a clear statement about the significance and novelty of our study, which is “Collectively, our study uncovers a novel biological function of aspirin in modulating the Drosophila antiviral immunity and provides theoretical bases for exploring new antiviral treatments in clinical trials” (Lines 23-25).

Consider rephrasing the sentence "Nevertheless, it remains probable that aspirin enables to execute other undescribed biological functions" to a more concise and direct statement.

    Response: According to the reviewer’s suggestion, we have modified the sentence into “Nevertheless, it remains probable that aspirin plays functional roles in other biological activities, for instance antiviral defense reaction” (Lines 16-17).

Introduction

Provide a more focused and concise introduction to the topic of antiviral drugs and the need for new compounds/treatments.

    Response: In the revised version of our manuscript (Lines 38-47), we have included in the first paragraph of the Introduction Section a more focused and concise description regarding antiviral drugs and the need for new compounds.

Consider reorganizing the information to present a logical flow of ideas and concepts.

    Response: In the present manuscript, the logic flow in the Introduction Section is as follows. First, we described the need and significance of exploring novel antiviral drugs. Second, we highlighted that Drosophila has been a well-established and cost-effective animal model for exploring antiviral drugs. Third, we demonstrated an antiviral role of the drug aspirin by using the Drosophila model. We believe that this logical flow is quite clear and easy for the readers to follow.

Provide a clear research gap and highlight the specific contribution of the current study in addressing this gap.

    Response: We have included a sentence describing the research gap in the Introduction Section (Lines 72-73). The last paragraph (Lines 77-83) points out the contribution of our study in addressing this gap.

Results

Subdivide the results section into subsections for clarity. Use more descriptive subheadings for each subsection.

    Response: In the Section of Results, we first described our findings that aspirin prevented viral replication in cultured S2 cells (Subsection 1). Second, we confirmed these observations in adult flies (Subsection 2). Third, we displayed that aspirin enhanced the survival of flies after viral infection (Subsection 3). These three Subsections examined and clearly demonstrated the anti-viral role of aspirin both in vitro and in vivo.

To illustrate the underlying molecular mechanism, we first paid our attention to the Imd protein, as we have already shown that this protein is one ubiquitination substrate of aspirin (Zhu et al., Aging and Disease, 2021, PMID: 34631223). However, loss of imd hardly impacted on the antiviral role of aspirin, implying an imd-independent manner in this case. We then performed a genetic screening on the major signaling pathways that control the fly antiviral reactions and found that loss of sting or relish reversed the antiviral function of aspirin (Subsection 4). Later, we presented evidence that aspirin ameliorates the expression of STING-induced genes (Subsection 5), indicating a potential involvement of the STING-NF-kB pathway in the regulatory intersection of aspirin in the fly antiviral defense. We deeply agree with this reviewer that the structure of the Section of Results should be reorganized. In the revised version of our manuscript, we have subdivided Subsections 4 into two separate Subsections. We believe that is a better way to present all the data in our study.

In addition, we have modified several subheadings as follows in a more descriptive manner.

Subheading 2.1 (Line 85): Additional aspirin leads to decreased viral titer in Drosophila S2 cells.

Subheading 2.2 (Line 108): Viral proliferation is limited in adult flies with dietary supplementation of aspirin.

Subheading 2.3 (Line 124): Dietary supplementation of aspirin enhances fly survival after viral infection.

Subheading 2.4 (Lines 146-147): Aspirin modulates fly survival and viral titer after infection in an imd-independent manner.

Subheading 2.5 (Lines 174-175): Aspirin relies on sting and relish to regulate fly survival and viral replication after infection.

Provide more context and explanation for the experimental methods used in S2 cell and fly experiments.

    Response: We have included more information in the Section of Materials and Methods (5.2, 5.3, and 5.4) as suggested.

Clarify the abbreviations "DCV", "CrPV", "VSV", "SINV", and "IIV6" and provide their full name upon first mention.

    Response: We have already provided the full names of these viruses when they were first described in the manuscript (Results 2.1).

Provide statistical analysis details and specify the significance thresholds for the results presented in Figure 1. Provide a more detailed explanation of the survival rate analyses presented in Figures 2D-2G. Provide statistical analysis details and specify the significance thresholds for the results presented in Figures 2 and 3.

    Response: We have modified the descriptions regarding statistical analyses in the Section of Materials and Methods (5.5) as suggested. The significance thresholds for each figure are shown in the corresponding figure legends.

Discussion

Discuss the potential mechanisms underlying the antiviral effects of aspirin in more detail.

    Response: In this study, we demonstrated the antiviral effect of aspirin both in vitro and in vivo. Of interest, we found that aspirin probably relied on the STING-NF-kB signaling pathway to regulate the fly antiviral defense. Our current evidence of course cannot fully illustrate how aspirin affects the STING signaling. However, based upon our experience and knowledge with regard to the molecular mechanism by which aspirin prevents Imd ubiquitination to delay fly aging, we would like to hypothesize that key factors of the STING pathway (for instance cGLRs, Sting, etc.) may be targeted by aspirin for ubiquitination regulation. We have included these pieces of information in the Discussion Section (Lines 234-248).

Compare and contrast the findings of the current study with previous studies on the antiviral effects of aspirin.

    Response: We have compared these findings in the Discussion Section (Lines 221-233).

Address the limitations of the study and suggest future directions for research.

    Response: According to the reviewer’s suggestions, we have included a statement regarding the limitations of our study and the future directions after the Discussion Section (Lines 261-266).

Overall

Improve the clarity and organization of the manuscript by rephrasing and restructuring sentences and paragraphs. Proofread the manuscript for grammatical and typographical errors. Ensure consistency in terminology and abbreviations throughout the manuscript.

    Response: We have reorganized the manuscript and gone through English editing with the help of a friend who is a native English speaker.

Reviewer 2 Report

Authors have demonstrated that aspirin benefits the fly antiviral defense in a STING signal-dependent manner in vitro and in vivo. In addition, they have studied how aspirin additionally ameliorates the expression of STING-induced genes that involved in stimulation of the interferon.

The phrase “As a result, the prior objective of this research is to confirm the antiviral effect of aspirin in vivo” should be tone down or add the clarifying information that it concerns fruit fly. 

Author Response

Authors have demonstrated that aspirin benefits the fly antiviral defense in a STING signal-dependent manner in vitro and in vivo. In addition, they have studied how aspirin additionally ameliorates the expression of STING-induced genes that involved in stimulation of the interferon. The phrase “As a result, the prior objective of this research is to confirm the antiviral effect of aspirin in vivo” should be tone down or add the clarifying information that it concerns fruit fly.

    Response: According to the reviewer’s suggestion, we have modified this phrase into “As a result, the prior objective of this research is to explore the potential involvement of aspirin in modulating the host antiviral reaction” (Lines 223-225).